# Preparation of a Dmap-Catalysis Lignin Epoxide and the Study of Its High Mechanical-Strength Epoxy Resins with High-Biomass Content

**DOI:** 10.3390/polym13050750

**Published:** 2021-02-28

**Authors:** Lingxia Song, Yeyun Meng, Peng Lv, Weiqu Liu, Hao Pang

**Affiliations:** 1Guangzhou Institute of Chemistry, Chinese Academy of Sciences, Guangzhou 510650, China; jnssllxx@163.com (L.S.); mengyeyun15@mails.ucas.ac.cn (Y.M.); lvpengtust@163.com (P.L.); 2University of Chinese Academy of Sciences, Beijing 100049, China; 3Guangdong Provincial Key Laboratory of Organic Polymer Materials for Electronics, Guangzhou 510650, China; 4CAS Engineering Laboratory for Special Fine Chemicals, Guangzhou 510650, China; 5CASH GCC(Nanxiong) Research Institute of New Materials Co., Ltd., Nanxiong 512400, China

**Keywords:** DMAP-lignin epoxide, bio-based epoxy resin, composite resin, tensile strength

## Abstract

The depletion of limited petroleum resources used for the fabrication of epoxy resins calls for the development of biomass-based epoxides as promising alternatives to petroleum-derived epoxides. However, it is challenging to obtain an epoxy resin with both high lignin content and excellent mechanical performance. Herein, a 4-dimethylaminopyridine (DMAP)-lignin epoxide with a certain epoxy value and a small molecular weight is obtained by the catalysis of DMAP for the macromolecular lignin. It was discovered that compared to the prepared composite resin of benzyltriethylammonium chloride (BTEAC)-lignin epoxide, there is a better low-temperature storage modulus for the DMAP-lignin epoxide resin and its composite resins with high-biomass contents, and higher tensile strength for its composite resins. In particular, the DMAP-lignin epoxide/ bisphenol A diglycidyl ether (BADGE) (DB) composite resin with DMAP-lignin epoxide replacement of 80 wt% BADGE, containing up to 58.0 wt% the lignin epoxide, exhibits the tensile strength of 76.3 ± 3.2 MPa. Its tensile strength is 110.2% of BTEAC-lignin epoxide/BADGE (BB) composite resins and is comparable to that of petroleum-based epoxy resins. There are good application prospects for the DB composite resin in the engineering plastics, functional composite, grouting, and other fields.

## 1. Introduction

As renewable biomass, lignin is abundant in nature and cheap to obtain. It is the second most abundant biopolymer after cellulose and the most abundant natural aromatic compound [1,2]. Nearly 50 million tons of lignin are produced each year, but only 2% are used for the production of value-added materials because of its complex construction and poor processability. With the efforts of researchers, lignin has been researched in many fields. The high carbon content in lignin can be used to prepare carbon-based supercapacitors and carbon fiber materials [3]. The phenolic hydroxyl group of lignin has a good reductivity for the synthesis of silver nanoparticles, and its macromolecular structure can be used as micro-carriers for the resulting silver nanoparticles [4]. Lignin has been transformed into polymers such as polyhydroxyalkanoates by means of microbial degradation [5,6]. Lignin as a macromolecular composed of benzene ring has been composited with petroleum-based compounds (such as resorcinol or phenol, urea, epoxy resins, and polyurethane), to improve the strength and toughness of composite resin and realize its application in the engineering field [7,8]. In particular, lignin, through chemical modification, can participate in the thermal curing reaction, which not only effectively improves the mechanical properties of other resins, but also realizes the partial and total replacement of the petroleum-based compounds, and solves the problems of the petroleum resource crisis and environmental pollution [9,10].

Lignin contains a large number of active hydroxyl groups and carboxyl groups, instead of bisphenol A to prepare the epoxy compound. The obtained lignin-based epoxy compound has thermosetting properties. However, compared with bisphenol A epoxy compound, the lower content of the epoxy group and the steric hindrance for the macromolecular lignin epoxides lead to poor mechanical properties for the resin, so it is very necessary to composite with bisphenol A epoxy compound. The lignin epoxide with the macromolecular structure is introduced into the bisphenol A epoxy resin matrix and participates in the chemical crosslinking of resin. As a result, it can effectively block the crack propagation fracture of bisphenol A epoxy resin and improve the strength and toughness of resin [1,11,12]. In addition to the above advantages, lignin epoxide itself will undergo certain changes of its chemical structure during continuous thermal processing, and condensation between the lignin fragments could occur, making the resin structure more packed to increase the strength [13,14,15]. Daniel Jason et al. synthesized the lignin-derived epoxy prepolymers via both the mild hydrogenolysis and the epoxidation of lignin and used it to replace 25–75% of the bisphenol A diglycidyl ether equivalent, discovering increases of up to 52% in the flexural modulus and up to 38% in the flexural strength [16]. Fatemeh Ferdosian et al. used the bio-based epoxy systems as polymer matrices for manufacturing fiber-reinforced plastics (FRPs) and coatings. The tensile and flexural strengths of the prepared FRPs using bio-based epoxy composites were superior or comparable to those of the FRP with the pure bisphenol A epoxy resin [17]. Claudio Gioia et al. investigated the relationship between thermomechanical properties and the chemical structure of the well-characterized lignin-based epoxy resins. It was discovered that compared to eucalyptus-based resins, the spruce-based lignin resin results in somewhat higher modulus and higher tensile strength because of the more condensed structures [18].

The above research work is mainly about the mechanical properties of the epoxy compound composite resin with the small molecule state lignin. Namely, The lignin is degraded or refined to a small molecular weight (M_w_ < 2000 Da). Then the lignin is epoxidized, and the composite resin is prepared. The epoxidation of lignin with higher molecular weight is rarely studied (M_w_ > 2000 Da). The main reason is that the intermolecular etherification crosslinking (crosslinking reaction between the epoxy group and hydroxyl group) easily occurs in the lignin epoxidation reaction, leading to an obvious increase in molecular weight and a severe decrease in compatibility with bisphenol A epoxide. Therefore, it is very important to control the molecular weight of lignin (M_w_ > 2000 Da) during the epoxidation and guarantee the epoxy group content.

The catalyst has a significant influence on the epoxy value and molecular weight of the reaction product [19,20]. For the catalysts, the selection is generally based on experimental design. There are two main methods: the one-step catalytic method of sodium hydroxide solution and the two-step catalytic method of ammonium salt-sodium hydroxide phase transfer [21,22]. The one-step method easily causes the etherification crosslinking reaction between molecules, resulting in the obvious increase of molecular weight. The two-step method, as a common method, can effectively promote the epoxidation of lignin and control the molecular weight effect. Herein, we found that 4-dimethylaminopyridine (DMAP) can be used to epoxidize lignin, with the catalytic effect comparable to that of quaternary ammonium salt catalyst, this can achieve the epoxidation of lignin and control the molecular weight of lignin epoxides. 

In this paper, DMAP-lignin epoxide was synthesized by catalysis of DMAP, and its molecular weight and epoxy value were evaluated. The mechanical properties of amine-cured DMAP-lignin epoxy resin and its composite resin with bisphenol A diglycidyl ether (BADGE) were studied. In order to better illustrate DMAP’s catalytic effect and its performance advantage as an alternative to the BADGE, the benzyltriethylammonium chloride (BTEAC)-lignin epoxide was compared with DMAP-lignin epoxide in terms of molecular features and resin properties.

## 2. Experimental

### 2.1. Materials and Chemicals

Lignin was purchased from Yanghai Environmental Protection material Co., Ltd. (Alibaba, Jinan, China). The lignin was from the straw plant. The parameters obtained for the purchased lignin are given in Table 1, and the typical structural units of lignin are described in Appendix A [23]. 

ECH (99.5%), NaOH (99.5%), and dimethyl sulfoxide (DMSO, 99.5%) were supplied by Aladdin Biochemical Technology Co., Ltd (Shanghai, China). DMAP (99.5%), triethylene teramine (TETA, 70.0%), BTEAC (99.5%), DMF (99.0%), and BDDGE (95.0%) were obtained from Shanghai Macklin Biochemical Co., Ltd. (Shanghai, China). BADGE (85%) was purchased from Sigma-Aldrich Trading Co., Ltd. (Shanghai, China). All of the chemicals were of analytical grade and used as received unless stated otherwise. 

### 2.2. Preparation of Lignin Epoxides

A given amount of lignin (4.80 g), DMAP (3.9 mmol, 0.48 g), and DMSO (50 mL) were placed in a round-bottom flask (250 mL) equipped with a thermometer, a stirrer, and a condenser. The mixture was heated to 117 °C under N_2_ bubbling conditions and was stirred for 30 min. Afterward, ECH (100 mL, 1.27 mol) was added while holding the reaction at 117 °C for 180 min. Then, the reactants were cooled to 70 °C before a NaOH aqueous solution (5 M, 18 mL) was added at the speed of 0.1 mL/min over a 180 min duration using a peristaltic pump. Subsequently, the reaction mixture was cooled to room temperature and was followed by transferring it to a beaker (500 mL). For the removal of the water-soluble impurities, including the catalyst, the deionized water (350 mL) was poured into the beaker, and intensely stirred. The mixture was allowed to stand for a sufficient length of time until a brown oily mixture formed at the bottom of the aqueous solution, then the water phase of the separated liquids was decanted. This process was repeated thrice. Next, dichloromethane (50 mL) was poured into the beaker under adequate agitation. The product was separated out of the mixture containing a small amount of residual water, which was then allowed to stand until the emergence of the layered liquids. The water and insoluble solid layers were discarded, and the deep brown dichloromethane solution, with the product dissolved, was obtained. In order to facilitate the next purification of the product, 30 mL of dichloromethane was removed from the dichloromethane solution by rotary evaporation. The dichloromethane solution was added drop-by-drop to ethyl acetate (200 mL) under magnetic agitation, and a brown precipitate was separated out of ethyl acetate. By filtering and blow-drying the brown precipitate at room temperature, the DMAP-lignin epoxide sample was finally obtained. The prepared sample was stored in DMF, with a solid content of 55–65%, to avoid an intermolecular crosslinking reaction that would decrease its solubility. 

The BTEAC-lignin epoxide was prepared according to a similar procedure as for the synthesis of the DMAP-lignin epoxide sample, using lignin (4.80 g), BTEAC (3.9 mmol, 0.89 g), DMSO (50 mL), ECH (100 mL, 1.27 mol), and NaOH (5 M, 18 mL) as the starting materials. 

### 2.3. Preparation of Epoxy Resins

#### 2.3.1. Preparation of the DMAP-Lignin Epoxide Resin, BTEAC-Lignin Epoxide Resin, and BADGE Resin

Into the DMF solution of the epoxide, a certain amount of the curing agent (i.e., TETA) was dropwise added while keeping agitation for homogeneous mixing at room temperature. The mixture was allowed to stand until the bubbles disappeared and was then cast into a polytetrafluoroethylene (PTFE)-lined mold to cure in an air-circulating oven. The following curing procedure was sequentially adopted: heating at 70 °C for 24 h, 120 °C for 4 h, and finally at 180 °C for 4 h. The epoxy resin sample was produced after demolding at room temperature. Different kinds of epoxy resin samples were prepared according to the recipe provided in Table 2. 

#### 2.3.2. Preparation of the Lignin Epoxide/BADGE Composite Epoxy Resin

The DMF solution of the lignin epoxide, BDDGE, and BADGE were successively dissolved in DMF solvent (3.5 mL); and the DMF solution with well-dissolved epoxides was obtained by adequate stirring. The amount of each added component is provided in Table 3. To promote the crosslinking and curing, we volatilized a certain amount of DMF, which was determined by the weight loss of the beaker loaded with the DMF solution during heating. After the reaction was continuously heated at 80 ^o^C until 3.0 mL DMF was volatilized, the heating was stopped, and the reaction mixture was naturally cooled to room temperature. A mixture of epoxides with good fluidity was obtained. After that, the amine-based curing agent (TETA) was homogenized to the mixture by stirring; and the sequential curing procedures were adopted. Specifically, the mixture was first cured at room temperature until it was immobile, followed by heating at 70 °C for 12 h, 120 °C for 8 h, and, finally, 150 °C for 6 h. The lignin epoxide/BADGE composite epoxy resin was obtained after demolding at room temperature.

### 2.4. Characterizations

Fourier transform infrared spectroscopy (FTIR) was adopted to analyze the functional groups of various samples using an infrared spectrometer (Bruker Tensor 27, Karlsruhe, Germany), with a scanning range of 400–4000 cm^−1^ and scanning resolution of 4 cm^−1^. The molecular structure of the prepared samples was also investigated using a ^1^H nuclear magnetic resonance (^1^H NMR) spectrometer (Avance III 400, Bruker Co., Ltd., Karlsruhe, Germany). For the ^1^H NMR test, the DMSO-d6 (0.6 mL) was adopted as the solvent, while *p*-nitrobenzaldehyde (*p*-NBD, 5.0 mg) was used as the internal standard. The mass of the specimen for the ^1^H NMR test was 25.0 mg. The locations of the ^1^H NMR peaks associated with the solvent were found at *δ_H_* 2.46–2.54 ppm, while those of the internal standard were observed at *δ_H_* 8.10–8.20 ppm, 8.30–8.50 ppm, and 10.10–10.20 ppm. 

### 2.5. Analysis of the Lignin Epoxidation 

Based on the ^1^H NMR spectra, the epoxy values were calculated by a comparison between the peak area of hydrogen arising from the epoxy group and the peak area indexed to the internal standard (*p*-NBD, *δ_H_* 10.10–10.20 ppm) [24]. The area in the chemical shift range of 2.60–2.90 ppm is the peak area of hydrogen arising from the epoxy group [25,26].

### 2.6. Molecular Weight Analysis 

Gel permeation chromatography (GPC) was employed to measure the molecular weight of the prepared samples using a Waters chromatograph (Milford, MA, USA) equipped with a Waters 2414 refractive index detector. The polymethyl methacrylate (PMMA) was adopted as the standard specimen and DMF as eluent. A DMF solution of the sample (3 mg/mL) was prepared, and a portion (50 uL) was withdrawn and was passed through a single PLgel MIXED-C column after being filtered with a 0.22 μm permeable membrane. The samples were then eluted using DMF at an elution rate of 1 mL/min and a column temperature of 35 °C. The molecular weight was finally measured by a comparison between the eluted sample and the standard sample. 

### 2.7. Test on the Mechanical, Thermomechanical, and Thermal Performance

Dynamic mechanical analysis (DMA) was performed using a DMA+300 analyzer (Metravib Co., Ltd., Limonest, France); and the storage modulus, loss modulus, and Tan δ were measured for the prepared samples. The single cantilever method was adopted for the DMA test on the sample with a size of 40.0 × 8.0 × 3.0 mm^3^, with a temperature ramp of −80 to 200 °C at a heating rate and vibration frequency of 3 °C/min and 1 Hz.

The calculation of the crosslinking density was carried out according to the following method: the temperature at the maximum of the peak on the curve of Tan δ vs. temperature was assigned to the glass transition temperature ($\;T_{g}$), and according to the classical rubber elasticity theory, the crosslinking density of the thermosetting epoxy resin was calculated by Equation (1) [1,27].
(1)$$\rho =E^{\prime }/3\mathrm{RT}$$
where *E′* represents the storage modulus of the sample at the temperature of $T_{g}+30\;℃$, and *R* and *T* denote gas constant and absolute temperature of the sample at the temperature of $T_{g}+30\;℃$, respectively. 

The tensile test was conducted using the 10 kN universal testing machine (Reger Co., Ltd., Shenzhen, China) according to the ASTM D638-2003 standard. The specimens were prepared in a dumbbell shape (with the width, thickness, and fixed length of 4.0, 2.0, and 25.0 mm, respectively), and the tension loading rate was set to 5 mm/min. Each sample was tested in a parallel fashion four times, and the average value was reported in this paper.

The fractured cross-sectional morphologies of the samples after finishing the tensile test were observed using an S-4800 scanning electron microscope (SEM, Hitachi Co., Ltd., Tokyo, Japan). Before the SEM observation, a thin layer of gold was splattered onto the sample surface. 

The thermal stability of the sample was evaluated using a TGA Q500 thermogravimetric analyzer (TA Co., Ltd., New Castle, DE, USA), with a temperature ramp of 30 to 800 °C at the heating rate of 10 °C/min. Thermal weight losses were measured under both N_2_ and air atmospheres at a gas flow rate of 50 mL/min.

## 3. Results and Discussion

### 3.1. Synthesis and Characterization of the Lignin Epoxides

Figure 1 shows the structure of the pristine lignin and the lignin epoxides examined by the FTIR spectrum. There are similar characteristic absorption bands before and after the epoxidation of lignin. The assignments for the peaks of lignin are listed in Table 4 [28]. The absorption band at 3450 cm^−1^ for the lignin, which can be assigned to the –OH stretching vibrations, becomes weakened in the DMAP-lignin epoxide, and the absorption peaks of C=O stretching at 1700 cm^−1^ and C–O stretching at 1260 cm^−1^ for lignin disappear. While the C=O stretching peak at 1710 cm^−1^ is attributed to the ester group and the C–O stretching peak at 1248 cm^−1^ representing the phenoxy group appear, and the absorption peak at 910 cm^−1^ appears corresponding to the epoxy group. The changes of active groups and appearance of the epoxy group are consistent with the characterization of Chizuru Sasaki et al. for the lignin epoxide [29]. These results confirm the introduction of the epoxy groups into the molecular structure of lignin as the result of an effective catalytic epoxidation reaction. A similar result can also be observed in the FTIR spectrum of the BTEAC-lignin epoxide sample, revealing that the catalysis of the lignin epoxidation can also be achieved with the BTEAC catalyst.

In order to further confirm the formation of the epoxy groups, the ^1^H NMR spectra of the DMAP-lignin epoxide and BTEAC-lignin epoxide samples are provided in Figure 2. The characteristic protons of the main groups of lignin are assigned in the regions of δ_H_ 3.50–4.00 ppm (CH_3_–O–), δ_H_ 4.01–4.89 ppm (–CH_2_–O–), δ_H_ 6.14–8.00 ppm (benzene ring), δ_H_ 8.61–9.80 ppm (phenolic hydroxyl group), and δ_H_ 12.04–13.00 ppm (–COOH). The signal peaks of carboxyl protons and phenolic protons disappear in the epoxidation lignin, and the signal peaks attributed to the epoxy group appear in the chemical shift range of δ_H_ 2.60–2.90 ppm and δ_H_ 3.20–3.50 ppm. These demonstrate the successful synthesis of the lignin epoxides. The earlier reports of Shou Zhao et al. support our results and show the same shift signal peaks of epoxy groups [25,26].

### 3.2. Evaluation of the Performance of the Prepared DMAP-Lignin Epoxide and Its Cured Resin

In addition to the epoxy value, the molecular weight and molecular weight distribution of the DMAP-lignin epoxide and BTEAC-lignin epoxide samples are also summarized in Table 5. Compared to the BTEAC-lignin epoxide, the DMAP-lignin epoxide possesses a lower molecular weight, narrower molecular weight distribution, and comparable epoxy value. Although DMAP and BTEAC have a similar epoxidized capability for lignin epoxidation, the DMAP catalyst can better control the molecular weight of the lignin epoxide as compared to BTEAC. The DMAP-lignin epoxide also exhibits better dispersibility and mobility during manufacturing processes that involve melting and curing.

The crosslinking density, storage modulus, and glass transition temperature (*T_g_*) of the three kinds of epoxy resins prepared by amine-based curing of the DMAP-lignin epoxide, BTEAC-lignin epoxide, and BADGE are summarized in Table 6. The order of the crosslinking density of these obtained resins is as follows: DMAP-lignin epoxide resin <BTEAC-lignin epoxide resin <BADGE resin. Such a sequence is consistent with the order of the epoxy value for the corresponding epoxides, namely the DMAP-lignin epoxide, BTEAC-lignin epoxide, and BADGE.

The variation of the storage modulus of the DMAP-lignin epoxide, BTEAC-lignin epoxide, and BADGE resin samples as a function of temperature is presented in Figure 3a. The storage modulus of the DMAP- and BTEAC-lignin epoxide resins are markedly lower than the BADGE resin at −50 °C, which is due to the much lower crosslinking density of the former (Table 6). When one compares the lignin-based resins, the DMAP-lignin epoxide resin shows a larger storage modulus as compared to the BTEAC-lignin epoxide resin, albeit with a lower crosslinking density. For the resin samples possessing similar crosslinking density, their packing structure can have a more significant impact on the storage modulus. The small molecular weight of DMAP-lignin epoxide resin can promote the packing density of resin (Appendix A), and further enhance the storage modulus in the glassy state. The influence of molecular weight on the dense packing and the storage modulus has been reported by Claudio Gioia et al., which is consistent with the conclusion in this paper [24].

The order of the *T_g_* of the three kinds of epoxy resins is as follows: BTEAC-lignin epoxide resin >BADGE resin >DMAP-lignin epoxide resin. All of *T_g_* is larger than room temperature, making these kinds of resins applicable at room temperature. The *T_g_* of DMAP-lignin epoxide resin is notably smaller than the BTEAC-lignin epoxide resin and BADGE resin due to its lower crosslink density [30]. Such a low *T_g_* of the DMAP-lignin epoxide resin makes it promising for toughening the petroleum-derived epoxy resin (e.g., BADGE resin). 

Although the BADGE resin has both high *T_g_* and high storage modulus (Figure 3a,b, and Table 6), it is still expected to achieve the co-curing composite of lignin epoxides with BADGE. For one thing, the lignin epoxide as an active macromolecule can strengthen the skeleton structure of BADGE resin by chemical reaction, and further improve the strength and toughness. For another, it is beneficial to promote the *T*_g_ and the crosslink strength of lignin epoxide resins and achieve a partial substitute for the BADGE.

In order to co-cure DMAP-lignin epoxide and BADGE, the compatibility between them should be the foremost problem to be resolved. The lignin epoxide is a polar macromolecular epoxide bearing hydroxyl groups, while BADGE is a viscous and aprotic polar epoxide with a low molecular weight. This discrepancy causes poor compatibility between them. In this study, the BDDGE diluent and the DMF solvent were employed to dissolve these two kinds of epoxides and improve their compatibility (see Experimental). After the lignin epoxide and BADGE had been dissolved, the DMF solvent was removed to facilitate the reaction between the epoxides and amine-based curing agent (in this case, TEAE). With the volatilization of the DMF solvent, the lignin epoxide was readily separated out of the BDDGE/DMF mixture because of its larger molecular weight. It is also worth noting that the higher content of DGEBA caused the lignin epoxide to be more easily separated out of the BDDGE/DMF mixed solution. Therefore, the lignin epoxide, with a larger molecular weight, exhibited a lower solubility during the manufacturing process with the lower content, resulting in the formation of the epoxy resin with obvious phase segregation. Unsatisfactory mechanical properties were subsequently measured. To ensure the good dissolution of the lignin epoxide before being cured, we added the TEAE for administering the curing reaction at room temperature once the DMF solvent was volatilized to a certain extent. The initial crosslinking reaction among the three kinds of epoxides, namely the lignin epoxide, BADGE, and BDDGE, resulted in the formation of an epoxy resin featuring a gel structure and integrated with a certain amount of the solvent that remained after the volatilization. Afterward, the solvent was further removed by heating and was accompanied by ring-opening polymerization. Finally, the epoxy resin was completely cured. Due to the lower molecular weight of the DMAP-lignin epoxide (as compared to that of the BTEAC-lignin epoxide), it was more resistant to be separated out of the BDDGE/DMF mixed solution, thus indicating its better solubility and processability.

The BADGE and DMAP- or BTEAC-lignin epoxide at a series of weight ratios were mixed into the BDDGE/DMF solution, resulting in the DMAP-lignin epoxide/BADGE (DB) and BTEAC-lignin epoxide/BADGE (BB) resins with various compositions. As you can see from Figure 4a, with the increase of weight of lignin epoxides, the tensile strength of composite resin first increased and then decreased. The tensile strength of DB60 was the highest, reaching up to 88.5 ± 0.8 MPa. When the lignin epoxides substituted 80 wt% of BADGE, the tensile strength of DB80 resin still remained 76.3 ± 3.2 MPa. At this point, the lignin epoxide content in the resin was up to 58.0%. The results showed that the composite resins with a high-biomass content still possess high tensile strength. This is because the DMAP-lignin epoxide participates in the crosslinking reaction of composite resin, changes the resin structure of composite resin without lignin epoxides (DB0), disperses the tensile stress effectively, and blocks the linear diffusion of cracks (Figure 5). Moreover, Continuous high curing temperature not only promotes complete crosslinking among the components but also enhances the binding of lignin epoxide itself. The high heating temperature is conducive to the increase of condensation between lignin fragments in composite resin, which in turn increases the bonding strength of the resin and improves the tensile strength of the resin. Although there have been reports on the effect of heating on the mechanical strength of lignin-based resins [13,14,15], the research is not systematic enough, and more studies on the heating effect on the properties of lignin-based resins should be conducted in the future. DB resin has better tensile strength than BB resin in the same mass ratio, which is due to its smaller molecular weight and narrower distribution of DMAP-lignin epoxide, resulting in better compatibility with BADGE and better homogeneity with each component of composite resin. With the ongoing rise of the lignin epoxide content, the elongation at break of DB resin first increased and then decreased similar to the change of its tensile strength, but the DB resin has the highest elongation at break with the 20% DMAP-lignin epoxide content (DB20) (Figure 4b). This result elucidates the fact that DB resin presents a better enhancement of toughening at lower biomass content.

After the tensile test, the fractured surface morphology of the DB resin was also studied by SEM observation; and the results are provided in Figure 5. No obvious phase interface can be observed in the composite resin, indicating that there is good homogeneity for each component of the composite resin. The DB0 without lignin epoxide is prone to form the brittle cracks of a radial pattern under the action of tensile stress (Figure 5a). When the lignin epoxide is added to the composite resin, the line crack propagation is blocked, forming a large number of regional areas with flabellate fracture surface morphology because of enhancement of the toughness (Figure 5b). When the content of the lignin epoxide compound is high, the crosslinking strength of the composite significantly decreases under the influence of the low epoxy value of lignin epoxide; and the tensile strength decreases accordingly, forming a fracture feature, like water-ripple morphology perpendicular to the tensile direction (Figure 5d). Only lignin epoxide and BADGE have a suitable mass ratio and uniform dispersion, and because the composite construct of their resin is stressed uniformly, the fracture surface morphology of the pretty turtle crack type will be formed. (Figure 5c). The BB resin presents a fractured surface morphology similar to the DB resin at the same weight percentage for lignin epoxide to BADGE (Appendix A).

Since the DB and BB resins were prepared by co-curing of the lignin epoxide, BADGE, and BDDGE, their storage moduli are mainly affected by the factors in terms of crosslinking density, phase structure, and the molecular weight and structure of the epoxides [11,24,31,32,33,34]. The DB80 resin shows a higher storage modulus compared to the DB40 resin at a low temperature (i.e., −50 °C), as shown in Figure 6a and Table 7. The crosslinking density can be decreased with the rise of the DMAP-lignin epoxide added amount; so the storage modulus should decrease. However, the DMAP-lignin-epoxide chain segment as the main component of DB80 resin, possessing a lower molecular weight, formed the more efficient packing of a dense 3D network structure with a smaller free volume. Moreover, the condensation reaction of the DMAP-lignin-epoxide chain segment in the curing by heating can also increase the compactness of the construct. This is why there is a 1461 MPa-larger storage modulus of the DB80 resin as compared to that of DB40 at −50 °C (Table 7). The low-temperature (−50 °C) storage modulus of the BB80 resin is close to that of its BB40 counterpart, but with a lower 260 MPa difference (Table 7). This is because the poorer stacking structure of BB composite resin has sacrificed the storage modulus. Compared to the DMAP-lignin epoxide resin, the DB80 resin shows 1.5- and 2.7-fold increased storage moduli at −50 and 25 °C, respectively. These results demonstrate that the DB80 resin with a high-biomass content has a good storage modulus at room temperature by the co-curing of DMAP-lignin epoxide and BADGE. 

The DB resins exhibit poor thermomechanical properties at temperatures higher than 100 °C, and no intact peak can be discerned from the curve of the tan δ vs. temperature, thereby making it difficult to obtain the *T_g_* (Figure 6b). Nevertheless, the *T_g_* values of the DB80 and DB40 samples are over 46 °C, which indicates that the co-curing DMAP-lignin epoxide with BADGE can improve the *T_g_* of the DMAP-lignin epoxide resin.

Since the multicomponent co-curing reaction was involved in the preparation of the DB resins, there are more than two characteristic peaks on the loss modulus curves. They mainly correspond to β and α peaks as generated by the motion of the lignin epoxide and BADGE chain segments, respectively (Figure 6c). The increase in the lignin epoxide content intensifies β peak and weakens α peak. Phase transition temperatures (namely T_α_ and T_β_) are mainly affected by the crosslinking construct of the DB resin, and the increase in the lignin epoxide content makes the T_α_ and T_β_ shift to lower temperatures (Table 7). The lower T_α_ and T_β_ of the DB resin (vs. BB resin) likely results from the lower molecular weight of the DMAP-lignin epoxide as compared to that of the BTEAC-lignin epoxide. 

The thermal stabilities of the DB and BB resins are further investigated, and the results are presented in Figure 7. Under an air atmosphere, the DB resin shows a variation tendency in the thermal degradation behavior similar to the BB resin (Figure 7a). There is a 5% weight loss at 250 °C for the DB resin; this is mainly due to the volatile components containing alkyl groups [35]. The thermal degradation occurring over the temperature range of 250–350 °C primarily arises from the alkyl ether chains. The thermal degradation over 250–350 °C begins to slow down, which can be associated with the alkyl groups connected to the phenyl oxygen occurring. With the ongoing temperature rise, the benzene ring structure starts to be thermally degraded, along with an increase in the thermal degradation rate. Finally, the DB resin is completely degraded at 650 °C [1,36]. Under the N_2_ atmosphere, a large number of residues remain at 650 °C for the DB resin (e.g., 30% and 20% masses that remain for the DB80 and DB40 resins, receptively), as shown in Figure 7b. The increase of the DMAP-lignin epoxide content leads to an increased quantity of the residues, which indicates that the residual chain segments of the DMAP-lignin epoxide are more resistant to the thermal degradation in the N_2_ atmosphere [37]. Under the conditions of the same lignin epoxide content, fewer residues can be found for the DB resin as compared to that for the BB resin. This result is likely due to the lower molecular weight of the DMAP-lignin epoxide composition, which probably causes the resulting DB resin to expose a large, thermally liable surface.

## 4. DMAP-Lignin Epoxide Perspectives and Challenges for Practical Application

In the face of the petroleum resource crisis and the awareness of ecological protection, it is very necessary to study and utilize lignin as a renewable, degradable, and environmentally-friendly polymer resource. Although there are many methods to degrade lignin into a small molecular state, the engineering utilization of macromolecular lignin is of great significance in reducing the cost of raw materials and enhancing the mechanical property of materials. DMAP catalyzes the epoxidation of macromolecular lignin, effectively controls the molecular weight of the epoxidation lignin, and then performs better in the molding process of the resin, forming a composite resin with high-biomass and high mechanical properties. The DMAP-lignin epoxide can be used in engineering plastics, functional composite, grouting, and other fields for development and utilization. However, it is still a big challenge to achieve the complete substitution of macromolecular lignin for petroleum-based raw materials to prepare the resin with strong mechanical properties and environmental friendliness. Because of the molecular steric hindrance of lignin epoxides, the bulk density of resin is affected; and the common solvents and curing agents will cause some environmental hazards. The control of both molecular weight and molecular structure of lignin should be further explored by the catalysts such as DMAP. At the same time, a green solvent such as ionic liquid should be developed for dissolving and curing the lignin epoxides and solving the existing environmental pollution problems.

## 5. Conclusions

This study has demonstrated that the DMAP is an active and useful catalyst for the epoxidation of the lignin biomass with ECH. The resulting DMAP-lignin epoxide has an epoxy value of as high as 2.09 mmol/g. Compared to the lignin epoxide produced with the BTEAC catalyst (corresponding to the BTEAC-lignin epoxide), the DMAP-lignin epoxide presents advantages in terms of structure, manufacturability, and performance of its resin. It has a smaller molecular weight, narrower molecular weight distribution, and better flowability during the manufacturing of the DMAP-lignin epoxide. Better low-temperature storage modulus and a more compact structure are shown for the corresponding DMAP-lignin epoxide resin. The DB composite epoxy resins with different biomass content have different performance advantages. The DB20 resin has the best toughness; the DB60 resin has the highest tensile strength, reaching up to 88.5 ± 0.8 MPa; the tensile strength of DB80 resin with 58.0 wt% lignin epoxide has 76.3 ± 3.2 Mpa, and the storage modulus has 4426 Mpa at 25 °C. The different types of composite resins meet the various mechanical performance requirements for practical applications. Therefore, such a low-cost biomass-based DMAP-lignin epoxide holds substantial potential for the partial substitution of the petroleum-based epoxy resin. 

## Figures and Tables

**Figure 1 polymers-13-00750-f001:**
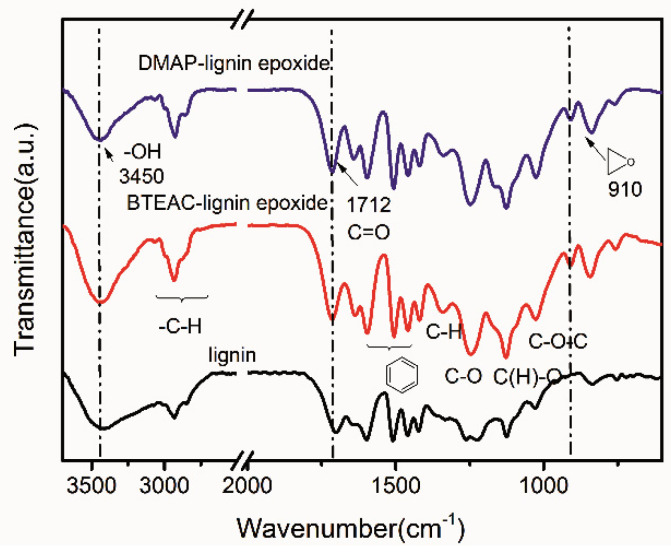
FTIR spectrum of the lignin, DMAP-lignin epoxide, and BTEAC-lignin epoxide.

**Figure 2 polymers-13-00750-f002:**
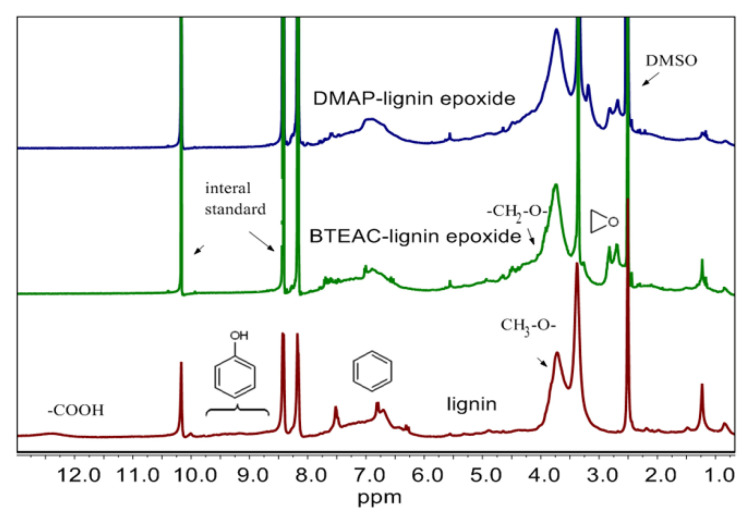
^1^H-NMR spectra of the lignin, DMAP-lignin epoxide, and BTEAC-lignin epoxide. *p*-NBD was adopted as the internal standard, with the characteristic ^1^H-NMR peaks at 8.10–8.20, 8.30–8.50, and 10.10–10.20 ppm.

**Figure 3 polymers-13-00750-f003:**
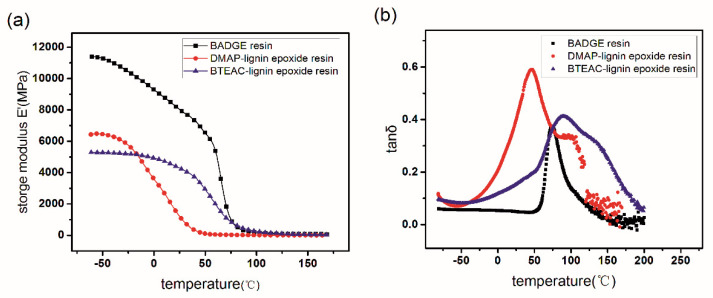
The storage modulus (**a**) and tan*δ* (**b**) measured for the DMAP-lignin epoxide, BTEAC-lignin epoxide, and BADGE resins as a function of temperature.

**Figure 4 polymers-13-00750-f004:**
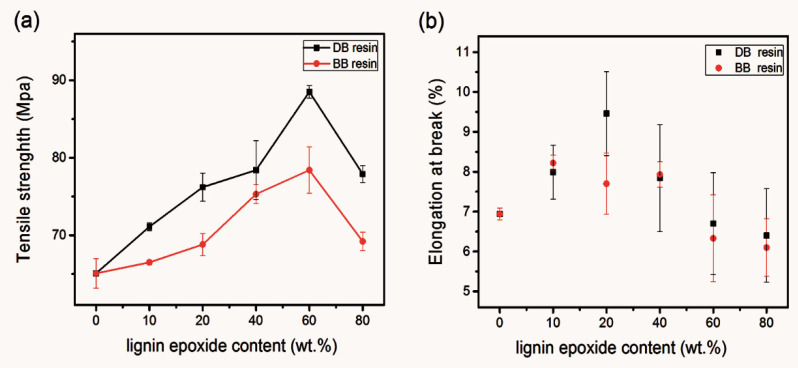
Tensile strength (**a**) and elongation at break (**b**) of the DB and BB resins with the different weight percentages of lignin epoxide. The lignin epoxide content is the weight percentages of DMAP-lignin epoxide replacement of BADGE.

**Figure 5 polymers-13-00750-f005:**
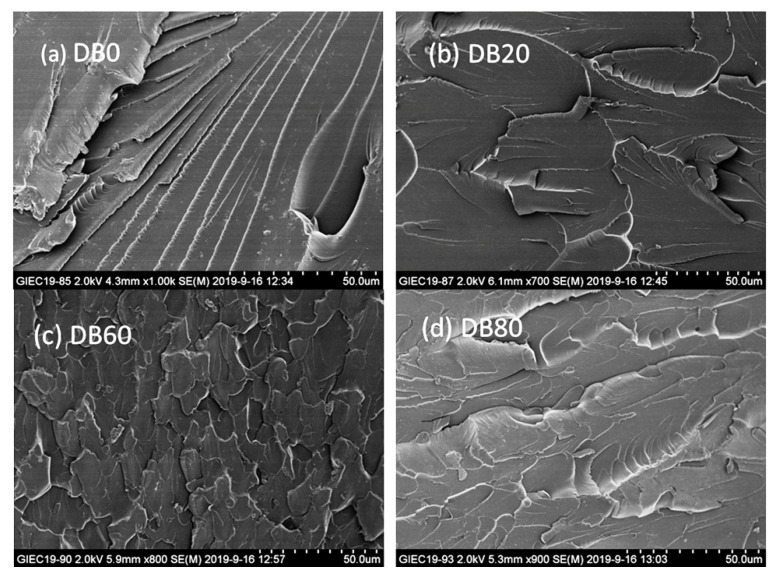
SEM images of the fractured surface of the DB0 (**a**), DB20 (**b**), DB60 (**c**) and DB80 (**d**) resin after tensile tests.

**Figure 6 polymers-13-00750-f006:**
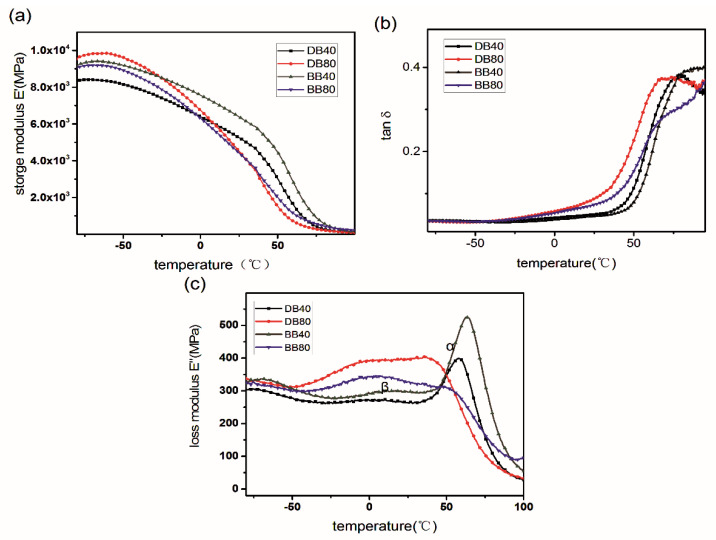
Plots of the storage modulus (**a**), tanδ (**b**), and the loss modulus (**c**) as a function of temperature for various DB and BB resin samples with different added amounts of lignin epoxide.

**Figure 7 polymers-13-00750-f007:**
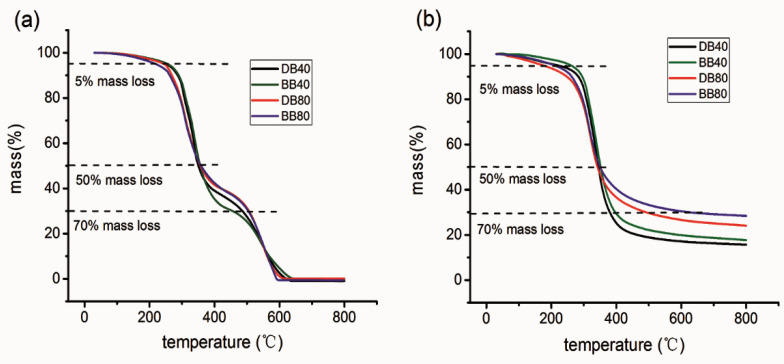
TG curves of the DB40, BB40, DB80, and BB80 resins under air (**a**) and N_2_ (**b**) conditions.

**Table 1 polymers-13-00750-t001:** The parameters obtained for the as-purchased lignin.

Type	M_n_	M_w_	Ɖ	ph–OH Content (mmol/g)	–COOH Content (mmol/g)	Ratio of G to S (G/S)
Softwood lignin	2827	3153	1.12	3.05	1.71	>50%

Note: M_n_ is the number-average molecular weight, M_w_ is the weight-average molecular weight, Ɖ represents molecular weight distribution, ph–OH denotes the phenolic hydroxyl group, –COOH is the carboxyl group, G and S represent guaiacyl and syringyl, respectively.

**Table 2 polymers-13-00750-t002:** The specifications of different types of epoxy resins: the 4-dimethylaminopyridine (DMAP)-lignin epoxide resin, benzyltriethylammonium chloride (BTEAC)-lignin epoxide resin, and bisphenol A diglycidyl ether (BADGE) resin.

Sample	DMAP-Lignin Epoxide	BTEAC-Lignin Epoxide	BADGE	TETA
	(g)	(g)	(g)	(g)
DMAP-lignin epoxide resin	22.00	-	-	3.00
BTEAC-lignin epoxide resin	-	22.00	-	3.00
BADGE resin	-	-	22.00	3.00

**Table 3 polymers-13-00750-t003:** The formulation adopted for the preparation of a series of the lignin epoxide/BADGE composite resin samples with different ratios of the lignin epoxide to BADGE.

Sample	DMAP-Lignin Epoxide	BTEAC-Lignin Epoxide	BADGE	BDDGE	TETA
	(g)	(g)	(g)	(g)	(g)
DB0/BB0	0.00	0.00	5.00	1.25	0.65
DB10	0.50	-	4.50	1.25	0.65
DB20	1.00	-	4.00	1.25	0.65
DB40	2.00	-	3.00	1.25	0.65
DB60	3.00	-	2.00	1.25	0.65
DB80	4.00	-	1.00	1.25	0.65
BB10	-	0.50	4.50	1.25	0.65
BB20	-	1.00	4.00	1.25	0.65
BB40	-	2.00	3.00	1.25	0.65
BB60	-	3.00	2.00	1.25	0.65
BB80	-	4.00	1.00	1.25	0.65

Note: DB10 represents the prepared DMAP-lignin epoxide/BADGE resin with 10 wt% DMAP-lignin epoxide and 90 wt% BADGE, respectively, while the other samples are named accordingly based on the weight percentage of lignin epoxide.

**Table 4 polymers-13-00750-t004:** Vibrational modes for FTIR absorbance peaks of lignin.

FTIR Band Position (cm^−1^)	Assignment
3450 cm^−1^	stretching of –OH in hydroxyl group and carboxyl group
3000, 2927, and 2846 cm^−1^	stretching of alkyl C–H
1700, and 1648 cm^−1^	stretching of C=O in carboxyl group
1600, 1506, and 1455 cm^−1^	stretching of C=C–C bond in benzene ring
1338 cm^−1^	symmetric deformation of C–H bond
1260 cm^−1^	stretching of C–O in phenol group
1222 cm^−1^	stretching of C–O in phenoxy group
1130 cm^−1^	stretching of C–O in alkyl alcohol
1030 cm^−1^	stretching of C–O in ether group
846, and 748 cm^−1^	deformation of C–H of benzene ring

**Table 5 polymers-13-00750-t005:** The epoxy value, molecular weight, and Ɖ of the lignin epoxides.

Samples	Epoxy Value (mmol/g)	Molecular Weight		
		M_n_	M_W_	Ɖ
DMAP-lignin epoxide	2.09	8244	17,085	2.07
BTEAC-lignin epoxide	2.16	9775	22,439	2.30

**Table 6 polymers-13-00750-t006:** Dynamic mechanical properties and crosslink density (*ρ*) of the DMAP-lignin epoxide resin, BTEAC-lignin epoxide resin, and BADGE resin.

Sample	Storage Modulus (MPa)			*T*_g_ (°C)	ρ $\left(10^{-3}\;\mathrm{mol}\cdot cm^{-3}\right)$
	Glassy Region at −50 °C	Glassy Region at 25 °C	Rubbery Region at *T*_g_ + 30 °C		
BADGE resin	11,300	8190	146.3	75	15.52
DMAP-lignin epoxide resin	6460	1200	28.4	46	3.26
BTEAC-lignin epoxide resin	5280	4286	43.8	90	4.47

**Table 7 polymers-13-00750-t007:** Storage modulus at the glassy region and transition temperatures for the DB40, DB80, BB40, and BB80 resins.

Sample	E′ at −50 °C	1st Transition T_β_ from E″	E′ at 25 °C	2nd Transition T_α_ from E″
	(MPa)	(°C)	(MPa)	(°C)
DB40	8168	4.8	5239	59.1
DB80	9629	−4.6	4426	37.6
BB40	9200	16.4	6336	62.9
BB80	8940	4.2	4350	52.5

## Data Availability

The data presented in this study are available on request from the corresponding author.

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
