# Peer review of "Preparation of a Dmap-Catalysis Lignin Epoxide and the Study of Its High Mechanical-Strength Epoxy Resins with High-Biomass Content"

_polymers, 2021, doi:10.3390/polym13050750_

Round 1
Reviewer 1 Report
An interesting manuscript, especially a good initiative to make green materials. Few points need to be considered as follows:
- Lignin used before with epoxy, polyurethane, and other polymers. Please shortly describe the previous work with those polymer/lignin in introduction.
- The manuscript has mainly three parts, synthesis and identification, characterization and properties. As the author mentioned '' high performance'' in title, this should be directly reflected on properties. The mechanical strength improved up to certain lignin content. However, it might not be good justification to say this material has high performance based on only mechanical strength. They can consider some other properties such as hydrophobicity, adhesive strength as well as a real practical application as a coating/adhesive or any other application.
Reviewer 2 Report
I read carefully an interesting research work entitled Preparation of a DMAP-catalysis Lignin Epoxide and the Study of its High-performance Epoxy Resins with High-biomass Content. Utilization of lignin to valuable products can be considered as a vital sustainable and eco friendly process.
Thus the concept of the manuscript fits and is suitable to publish in Polymers Journal. This manuscript is generally well written and clearly presented however some comments should consider to improve then quality of the manuscript
- The main lacking part of this manuscript is very little discussion of results with the previous results of literature needing substantial discussion at revision stage.
- In abstract authors should mention the importance of research work in one or two sentences.
- Provide a nice graphical abstract representing the overview of the MS with key highlights. In the manuscript many abbreviations are used so add all abbreviations and their full form after the abstract section would be better.
- In the introduction section, write the novelty of the work and the problem statement clearly. Avoid clusters of references and give details for each reference. Authors should discuss some recent applications of lignin for NPs synthesis and biopolymers production pl refer an cite International journal of biological macromolecules 128, 391-40, 2019; Bioresource Technology Volume 325, April 2021, 124685. The detailed discussion about the novelty, significance of your research work and research gap relative to the literature is essential.
- Table S1 should be in the main text.
- Statistical analysis of the results should be provided in the materials and methods section. It's important for all experimental work Report these values in the results and discussion.
- In figure and table always give full form of abbreviation. In addition, the figure and table caption give all details.
- FTIR and NMR analysis give details of all peaks and also compare the results with the literature in brief.
- Write the practical applications and future research perspectives and challenges by adding a new section before conclusions.
- What are the limitations to use this methodology for commercial application?
- The conclusion of the study is not discussed with the specific output obtained from the study, it could be modified with precise outcomes with a take home message.
- English and grammar mistakes are present. The author should check the manuscript by native English Speaker to improve the quality of the manuscript.
Round 2
Reviewer 2 Report
The authors have substantially revised the manuscript according to the comments.
The present form of the manuscript can be accepted for publication.
Author Response
Dear Reviewers
Thank you very much for your comments. We would like to sincerely thank you for your time and efforts in this article again. Best wishes to you!
Yours sincerely,
Hao Pang
Professor, Guangzhou Institute of Chemistry,
Chinese Academy of Sciences
Guangzhou, Guangdong, China, 510650
E-mail: panghao@gic.ac.cn, Tel :+86 18028632718.